# In Vitro Validation of Phosphorodiamidate Morpholino Oligomers

**DOI:** 10.3390/molecules24162922

**Published:** 2019-08-12

**Authors:** May T. Aung-Htut, Craig S. McIntosh, Kristin A. West, Sue Fletcher, Steve D. Wilton

**Affiliations:** 1Centre for Molecular Medicine and Innovative Therapeutics, Murdoch University, Perth, WA 6150, Australia; 2Perron Institute for Neurological and Translational Science, the University of Western Australia, Perth, WA 6009, Australia

**Keywords:** antisense oligonucleotide, morpholino, PMO, transfection, electroporation, exon skipping

## Abstract

One of the crucial aspects of screening antisense oligonucleotides destined for therapeutic application is confidence that the antisense oligomer is delivered efficiently into cultured cells. Efficient delivery is particularly vital for antisense phosphorodiamidate morpholino oligomers, which have a neutral backbone, and are known to show poor gymnotic uptake. Here, we report several methods to deliver these oligomers into cultured cells. Although 4D-Nucleofector™ or Neon™ electroporation systems provide efficient delivery and use lower amounts of phosphorodiamidate morpholino oligomer, both systems are costly. We show that some readily available transfection reagents can be used to deliver phosphorodiamidate morpholino oligomers as efficiently as the electroporation systems. Among the transfection reagents tested, we recommend Lipofectamine 3000™ for delivering phosphorodiamidate morpholino oligomers into fibroblasts and Lipofectamine 3000™ or Lipofectamine 2000™ for myoblasts/myotubes. We also provide optimal programs for nucleofection into various cell lines using the P3 Primary Cell 4D-Nucleofector™ X Kit (Lonza), as well as antisense oligomers that redirect expression of ubiquitously expressed genes that may be used as positive treatments for human and murine cell transfections.

## 1. Introduction

Antisense oligonucleotides (ASOs) are short, single-stranded molecules that can be designed to specifically bind to DNA or RNA via Watson-Crick base pairing, with the aim of modifying specific gene expression. Although ASOs have been used as laboratory tools since these agents were first reported, it took more than three decades for their arrival in the clinic as therapeutics. An extensive amount of time and effort can often be spent on in vitro development, assessment, and validation of antisense oligomer efficacy, to ensure that the optimal ASOs progress to clinical trials. In 1978, Zamecnik and Stephenson were the first to report blocking Rous sarcoma viral replication in chick embryo fibroblast cell cultures after transfection with a 13-nucleotide oligomer and, despite not knowing the mode of action, proposed the use of ASOs as antiviral agents [1,2]. As natural DNA oligonucleotides with the phosphodiester backbone are susceptible to enzymatic degradation, chemical modifications to bases and backbones were developed to enhance binding affinity, increase resistance to nuclease digestion and allow different modes of action that include specific degradation through RNase H induction or siRNA action, blocking protein translation or redirecting pre-mRNA processing [3,4,5,6]. Since these modifications and mechanisms were introduced, research into ASO compounds as therapeutics for many diseases, including those caused by genetic mutations, has grown immensely. 

The U.S. Food and Drug Administration has now approved several ASO drugs involving various chemistries in different stages of clinical development [7,8,9,10]. The success of all approved therapeutic ASOs was based upon careful design and screening of the optimal sequences, first in an in vitro system and then in vivo. Therefore, it is imperative to have confidence in a consistent and reliable delivery protocol to ensure cellular or nuclear uptake of the ASOs [11]. 

Unlike many other synthetic nucleic acid analogues in clinical development, phosphorodiamidate morpholino oligomers (PMOs) have a neutral charge and delivering these compounds into cultured cells can be problematic [5,12,13]. Although gymnotic delivery for some PMOs has been reported in human myotube cultures when applied at high concentrations [14], this process is largely inefficient. The inefficiency is generally attributed to the PMO’s inability to translocate through cell membranes as an uncharged molecule and may be overcome by the addition of a peptide-conjugated tag, which provides a significant charge to the PMO allowing for easier delivery. Thus, uncharged PMOs generally require delivery agents or protocols to enhance intracellular uptake, although recently Miyatake et al. (2019) demonstrated that scavenger Receptor Class A1 mediated uptake of PMOs both in vitro and in vivo [15]. This study may provide further insight into the gymnotic uptake of these compounds and lead towards an improved drug delivery system. 

One must be confident that any transfection protocol used efficiently delivers the PMO into the cell. As learnt from previous reports, in vitro analysis of DMD exon 46 skipping using a morpholino oligomer was ineffective compared to the same sequence composed of 2′*O*-methyl-modified bases on a phosphorothioate backbone (2′-OMe ASOs) [16]. Conversely, in vivo assessment of PMOs showed efficient exon 46 skipping of the DMD transcript in humanised DMD mice [17]. Therefore, the discrepancy between these results arose from an issue of delivery rather than an ineffective ASO, highlighting the need for confidence in efficient delivery in vitro during ASO development.

The importance of PMO chemistry for protein studies was revealed during the development of ASOs to correct the defect in the canine model of DMD [18]. Antisense oligomer optimisation to skip dystrophin exons 6 and 8 used the 2′-OMe ASOs, and while this chemistry was suitable for RNA studies, detecting dystrophin restoration was not possible [18]. However, when the equivalent ASOs were prepared as PMOs or PMOs conjugated to a peptide-conjugated tag (PPMO), dystrophin protein of near full-length was readily detected [18]. When the uncomplexed PMO cocktail was added to cell media, uptake was poor and dystrophin expression modest. In contrast, the PPMO-treated canine DMD cells showed robust multi-exon skipping and unequivocal dystrophin restoration [18]. 

One simple and relatively effective means of delivering PMOs into cultured cells is “scrape-loading”, where adherent cells are gently scraped to create small transient holes in the plasma membrane, thus allowing for better uptake of the PMO [19]. Alternatively, Gebski et al. reported annealing sense-strand DNA “leashes” to antisense PMOs, thereby allowing the negatively charged PMO: DNA duplex to form complexes with commercially available cationic lipid transfection reagents [20]. In this approach, the PMO delivery is more comparable to that used to transfect the commonly used ASOs composed of 2′-OMe or locked nucleic acid (LNA) oligomers. Gene Tools has developed a novel peptide-mediated delivery reagent, Endo-Porter, that allows PMO delivery via endocytosis to adhered cells without the use of a leash [21]. Electroporation-mediated transfection using a 4D-Nucleofector™ or Neon™ system also offers efficient delivery of PMOs to different cell types, including both adherent and suspension cells [22,23].

The first compound to be granted accelerated approval for the treatment of Duchenne muscular dystrophy, *Exondys 51*, was developed in our laboratory [24,25,26], and we are currently researching nucleic acid therapeutics for numerous diseases amendable to RNA splicing intervention [24,25,27,28,29]. Consequently, we have experience in the design, delivery and evaluation of hundreds of PMOs in different cell lines to address various mutations. We found that PMOs can be delivered into adherent cells, without a leash, using commercially available cationic lipid-mediated transfection reagents, with an efficiency similar to that observed with nucleofection. Cationic lipid transfection reagents were chosen due to the ease of accessibility to most laboratories throughout the world. In addition, little literature exists regarding alternative transfection methods. Of the transfection reagents evaluated, we found that Lipofectamine 3000™ had the least impact on cell morphology and viability but offered relatively effective PMO delivery into fibroblasts. In addition to Lipofectamine 3000™, Lipofectamine 2000™ also offered effective PMO delivery into cultured myoblasts/myotubes. Nucleofector or Neon electroporator instruments and their consumables are relatively costly; however, the advantage of these delivery systems is that the PMO can be delivered to as many as 20 or 6 million cells, respectively, using substantially lower amounts of PMO. Consequently, we routinely use nucleofection for PMO delivery, and we report optimised PMO delivery programs using the P3 Primary Cell 4D-Nucleofector™ X Kit (Lonza) that can be used for various cell lines. 

## 2. Results

### 2.1. PMO Delivery into Human Dermal Fibroblasts

We have chosen previously optimised exon skipping PMOs targeting the *ITGA4* and *SMN* transcripts expressed in most cell types and cell lines (Appendix A). To determine whether the nominated widely used cationic lipid transfection reagents available in the laboratory offer an economical and reliable method to deliver PMOs into human dermal fibroblasts, we selected two commonly used cationic lipid transfection reagents; Lipofectamine 3000™ and Lipofectin™, for complex formation with ASO 1, optimised to induce skipping of exons 3 and 4 from the *ITGA4* pre mRNA transcript [30]. In addition to the lipoplex: PMO transfection, alternative transfection strategies including nucleofection, Endo-Porter and gymnotic uptake of uncomplexed PMO were evaluated in a direct comparison. The same experiments were repeated using human myoblasts, except Lipofectin™ was replaced with Lipofectamine 2000™ for comparison, as the latter shows better transfection efficiency in myoblasts in our laboratory. 

The morphology and viability of fibroblasts and myoblasts treated with PMOs complexed with the Lipofectamine 3000™ reagent were similar to that of cells treated with the uncomplexed PMO and untreated fibroblasts or myoblasts (Figure 1A,B). In contrast, substantial cell death was observed after fibroblasts were treated with PMO complexed with Lipofectin™ (10 µl/mL) and myoblasts treated with PMO: Lipofectamine 2000™ complex (Figure 1C,D). Nucleofection did not visibly affect the morphology or viability of the fibroblasts or myoblasts (Figure 1A–D). The morphology of fibroblast cultures after Endo-Porter-mediated PMO delivery was atypical when compared to other reagents (Figure 1A). Aggregates, possibly complexes of PMO and peptide, bordered the cells, and small cell bodies with slender cell extensions were prominent in Endo-porter treated cells, but not observed in untreated fibroblasts. 

The uptake of ASO 1 and subsequent ability to redirect pre-mRNA processing was demonstrated by RT-PCR analysis of the *ITGA4* transcripts (Figure 1E,F). All treatments, except for the uncomplexed PMO, induced exon 3 and 4 skipping with similar efficiencies, although higher levels of exon skipping were observed in the fibroblasts treated with PMO: Lipofectin. Increased levels of exon skipping may have been a consequence of the extensive cell death observed after the Lipofectin™ transfection, compared to the other treatments. Similarly, myoblasts treated with PMO: Lipofectamine 2000™ caused more than 50% cell death. The average percentages of exon skipping from three biological replicates are shown in Figure 1E,F and the original gel images are shown in Appendix A. 

Although evidence of cell death appeared minimal after Endo-Porter-mediated PMO delivery (Figure 1C,D), the fibroblasts nevertheless showed signs of cell stress. Therefore, we analysed the expression profile of the cell stress-related proteins using the proteome profiler human cell stress array kit (Appendix A), which consists of an analysis of relative expression levels of 26 cell stress-related proteins. Although the levels of some of the stress-related proteins were altered, the most compelling changes were observed in two proteins, Cited 2 and p21, which were decreased by more than 50% in Endo-Porter treated fibroblasts, compared to the untreated cells.

### 2.2. Nucleofection Programs for PMO Delivery into Different Cell Lines

Nucleofection is one of the commonly used methods to efficiently deliver nucleic acids, including PMOs, into cultured cells. Nucleofection is our preferred method for PMO delivery as the test compounds can be delivered into a million cells (using the 4D-Nucleofector X Kit S) without compromising the efficiency (Figure 2A) or using excessive amounts of PMO, as well as being highly reproducible (Figure 2A). We have optimised nucleofection programs using the P3 kit (Table 1) for multiple, extensively used cell lines that are commercially available, and achieved 70% or more exon skipping of the target transcript with less than 10% cell death as shown in Figure 2B. We selected two PMOs, ASO 2 targeting exon 3 of the *ITGA4* transcript (Figure 2A), and ASO 3 targeting exon 7 of the murine *Smn* transcript [31] for standardised evaluation (Figure 2B). Although ASO 3 was designed to target the murine *Smn* transcript, it is also capable of inducing exon 7 skipping from the human *SMN* transcript despite 2 mismatches at the 8th and 21st position. From these experiments, it can be seen that nucleofection is an efficient and reproducible method of transfection.

## 3. Discussion

It is often stated that there are three great challenges to genetic therapies: delivery, delivery and delivery. While the design of the therapeutic vector, plasmid or antisense oligomer is a fundamentally crucial aspect, without efficient and effective delivery to the target tissue or cells, the most therapeutically active compound may appear inert. We have extensive experience in designing and assessing the potential of splice modulating antisense oligomers, initially as research-grade 2′-OMe phosphorothioate oligonucleotides that can be readily synthesised in-house, and perhaps more importantly, easily and efficiently transfected into cultured cells as cationic lipoplexes. 

Apart from the considerable cost, PMOs have other limitations, including poor cellular/nuclear uptake unless particular strategies are employed. Indeed, PMOs were discounted as possible splice switching agents for dystrophin exon skipping after researchers compared PMOs, 2′-OMe, LNA and peptide nucleic acids (PNA) [16]. The PNA was ineffective while the PMOs induced only weak exon skipping compared to the 2′-OMe ASOs and LNAs. It is of interest to note that the transfection conditions to compare these different oligonucleotides were not equivalent in this study, with polyethylenimine complexed with the 2′-OMe and LNA oligomers, ethoxylated polyethylenimine mixed with PMO and a DNA sense strand, and no transfection agent for the PNA. We had earlier shown that an optimised PMO, when annealed to a variety of sense strand DNA/RNA-like leashes and complexed with Lipofectin™, was found to induce robust exon skipping after transfection at concentrations as low as 30 nM, whereas the uncomplexed PMO showed no activity at 30-fold higher transfection concentrations [20].

Conscious of the importance of PMO delivery, we evaluated various protocols to introduce PMOs into cultured cells. The most common cell types studied in our laboratory are human dermal fibroblasts and myoblasts, as these cells could be readily obtained from individuals after Human Ethics Committee approval and informed consent through our collaborating clinicians. Depending upon commercial availability and tissue-specific gene expression, other cell lines may also be studied, and we provide nucleofection programs for the P3 kit to deliver PMO into various cell lines, in addition to liposome transfection. 

Convention dictates that when delivering uncharged PMOs into cells with cationic liposome transfection reagents, annealing the PMO to a complementary strand DNA or RNA-like “leash” would provide the necessary negative charge for PMO: lipoplex formation. Counter-intuitively, we found that a leash was not necessary for PMO: lipoplex formation with either Lipofectamine 3000™/2000™ or Lipofectin™. Significantly, these reagents can deliver PMOs into cells with efficiencies similar to that achieved by nucleofection. Transfection with ASO 1 showed consistent but modest levels of exon skipping and was therefore chosen for study so it would be possible to discern between the efficiencies of different delivery protocols confidently [31]. When assessing a PMO that induces efficient skipping (>80%), it can become more difficult to reliably discern subtle differences in transfection efficiency and densitometric analysis with titration experiments. Uncomplexed ASO 1 was not taken up by fibroblasts after 24 h incubation, and therefore, any exon skipping detected in the fibroblasts transfected with ASO 1 complexed with transfection reagent must be attributed to enhanced delivery through these transfection reagents. In our experience, Lipofectamine 3000™ is versatile and shows little or no toxicity in most cell types. It is possible that the cell death observed in fibroblasts treated with Lipofectin™ may be limited by lowering the concentration of Lipofectin™ utilised for PMO delivery. However, the 10 µL/mL Lipofectin™ we applied is lower than the recommended DNA: transfection reagent ratio of 1:1. Lipofectamine 2000™ consistently outperformed Lipofectamine 3000™ in myoblasts regarding the exon skipping efficiency, although it caused higher levels of cell death. Among the biological replicates performed in myoblasts, we noticed that there was a positive correlation between the extent of myotube formation and uncomplexed PMO uptake. However, once the PMO was complexed with a delivery agent, uptake improved in all cultures, regardless of myotube formation. 

The fibroblasts treated with Endo-Porter and PMOs showed anomalies around the cell membrane, and this may be due to the novel peptide promoting endocytosis and thus stressing the membrane. Although endocytosis is a natural biological process in most cells, we speculate that Endo-Porter may induce a degree of endocytosis that exceeds the capacity of the pathway during normal endogenous function, compromising the cell membrane and thus inducing cell stress. Analysis of cell stress-related protein levels in Endo-Porter-treated cells suggested that cells may be under considerable stress, as indicated by the reduced expression of Cited 2, one of the proteins downregulated under ER stress [32]. Although the level of p21 in Endo-Porter-treated fibroblasts was lower than that in untreated fibroblasts, we speculate that the untreated fibroblasts were more confluent at collection than the Endo-Porter treated fibroblasts, accounting for the relatively higher p21 level.

In summary, this study provides various PMOs to modify the expression of broadly expressed genes that may be used as positive treatments for assessment of PMO delivery in a range of cultured cell types. We recommend the use of an electroporation system, particularly nucleofection that, over the years, has provided efficient and reproducible PMO delivery into many different cell types in our laboratory, with very little evidence of stress on the cells. We have provided optimised programs for various cell lines using a single P3 kit, but if this system is not available, Lipofectamine 3000™ can be used for reliably delivering PMO to most cell types, including fibroblasts, and Lipofectamine 2000™ for myoblasts. The PMOs described here target genes widely expressed in many different cell types and could be used as controls to monitor and confirm efficient delivery into most cell lines under investigation.

## 4. Materials and Methods

### 4.1. PMO Nomenclature

The nomenclature of all PMOs outlined in this study is as described by Mann et al. 2002 [33], as it indicates annealing coordinates and allows a sharp distinction between overlapping ASOs targeting a common region. Briefly, the first letter designates the species (H: human) followed by the targeted exon number with the specification of an acceptor (A) or donor (D) site and the annealing coordinates in brackets from 5′ to 3′ position of the mRNA transcript. The intronic bases are represented by negative (−) and the exonic position by positive (+). The annealing coordinates were based on the reference transcript as denoted by *NCBI* and *Ensembl genome browser 96*.

### 4.2. PMOs

The PMOs used in this study, optimised after microwalking and designed to redirect *ITGA4* or *Smn* pre-mRNA processing [30,31] (Table 2), were purchased from Gene Tools, LLC (Philomath, OR, USA). These oligomers target ubiquitously expressed genes (*ITGA4* and *Smn*) and may be useful as positive transfection controls in other studies. These gene targets were chosen as they are widely expressed throughout common commercial cell lines. Expression patterns of both genes (adapted from GeneCards.org) can be found in Appendix A.

### 4.3. Cell Culture 

All cell culture reagents were purchased from Thermo Fisher Scientific Australia Pty. Ltd. (Scoresby, VIC, Australia) and cultures were maintained at 37 °C under a 5% CO_2_/95% air atmosphere, unless otherwise stated. The use of human cells was approved by the Murdoch University Human Research Ethics Committee (approval 2013/156). Human dermal fibroblasts were propagated in DMEM supplemented with L-Glutamine and 10% foetal bovine serum (FBS). Human myogenic cells were prepared from biopsies taken from healthy individuals undergoing elective surgery at Royal Perth Hospital, Perth Western Australia, as described by Rando and Blau [34] with minor modifications [35]. Primary human myogenic cells were propagated in Hams F-10 Medium supplemented with 20% FBS and 0.5% chick embryo extract (Jomar Life Research, Scoresby, VIC, Australia) on flasks coated with 100 µg/mL Matrigel (BD Biosciences, Sydney, NSW, Australia). Jurkat cells were supplied by the European Collection of Cell Cultures (ECACC; Salisbury UK) and purchased from CellBank Australia (Westmead, NSW, Australia) and maintained in RPMI-1640 supplemented with 10% FBS. Human glial (oligodendrocytic) hybrid cell line MO3.13 was purchased from BioScientific Pty. Ltd. (Kirrawee, NSW, Australia) and maintained in DMEM supplemented with 10% FBS. The human bone marrow neuroblastoma cell line, SH-SY5Y, was purchased from ATCC (In Vitro Technologies Pty. Ltd., Noble Park North, VIC, Australia) and maintained in a 1:1 mixture of MEM and F-12 Medium supplemented with 10% FBS. The human hepatocarcinoma cell lines, Huh7 and HepG2, were supplied by the JCRB Cell Bank (Osaka, Japan) and purchased from CellBank Australia (Westmead, NSW, Australia). These cells were maintained in DMEM supplemented with 10% FBS. Murine H2k *mdx* myoblasts were cultured in poly D-lysine (50 µg/mL) and Matrigel (100 µg/mL) coated flasks at 33 °C under a 10% CO_2_/90% air atmosphere in high-glucose DMEM supplemented with 20% FBS and 10% horse serum, 0.5% chicken embryo extract and 20 units/mL γ-interferon (Roche Products Pty. Ltd., Sydney, NSW, Australia). 

### 4.4. Isolation of Splenocytes 

Splenocytes were freshly isolated from the spleens taken from two *mdx* mice (Murdoch University approval number R2829/16). All steps were performed on ice and in RPMI-1640 supplemented with 10% FBS. Both spleens were sliced into small pieces (approximately 4 mm^2^) and mashed using the frosted side of pre-sterilised glass slide. The undissociated pieces of spleen were removed by centrifugation at 100× *g* for 8 min before the supernatant was centrifuged for a further 8 min at 200× *g*. The pellet of splenocytes was resuspended in 10% FBS, RPMI-1640 and the number of cells was determined using a haemocytometer (Sigma-Aldrich, Sydney, NSW, Australia).

### 4.5. Transfection

Approximately 70% confluency/15,000 fibroblasts per well were plated in a 24-well plate one day prior to transfection [31]. Human myoblasts were plated in a 24-well plate that had been previously coated for 1 h with 50 µg/mL poly D-lysine (Sigma-Aldrich, Sydney, NSW, Australia) and 100 µg/mL Matrigel at 70% confluency/30,000 cells per well in Low Glucose DMEM supplemented with 5% horse serum, and cultured for 24 h [36]. Transfection complexes were formed according to the manufacturer’s recommendations. PMO was transfected at a final concentration of 10 µM using 3 µL/mL Lipofectamine 3000™, 10 µL/mL Lipofectin™ or 10 µL/mL Lipofectamine 2000™. Briefly, the PMO and transfection reagents were separately diluted into 50 µL OptiMEM before mixing. PMO/lipid complexes were formed by incubating for the recommended time (10–15 min Lipofectamine 3000™; 30–45 min Lipofectin™; 5 min Lipofectamine 2000™) at room temperature before topping up to 300 µL with OptiMEM and adding to the cells. For Endo-Porter (Gene Tools) transfection, 80% confluency/25,000 cells were plated per well (as per the manufacturer’s recommendations) one day before transfection, replaced with fresh media, and PMO was added to the media before adding the Endo-Porter (3 µL/mL).

### 4.6. Nucleofection

All nucleofections were performed using the 4D-Nucleofector™ X Unit and P3 kit (Lonza, Mt Waverley, VIC, Australia) with the concentration of PMO set at 50 µM in the cuvette (volume of 1 µL 1 mM PMO in 20 µL cuvette volume), unless otherwise stated. The cell type, number of cells, and the program used for nucleofection is shown in Table 1, in accordance with the manufacturer’s recommendations. After the nucleofection pulse with the PMO, the cells were allowed to recover for 10 min at room temperature before being resuspended in the appropriate growth medium. Total RNA was extracted after 24 h using Direct-zol™ RNA Kit (Zymo Research, Tustin, CA, USA) according to the manufacturer’s instructions. 

### 4.7. Microscopy

The images of transfected cells were captured using a Nikon^®^ TS100 (Nikon, Sydney, NSW, Australia) prior to extracting total RNA using the Direct-zol™ RNA Kit.

### 4.8. RT-PCR

RT-PCR was performed using a Superscript III One-Step RT-PCR System (Life Technologies, (Scoresby, VIC, Australia). Total RNA (50 ng) from PMO-treated and untreated cells was used as a template for all reactions. RT-PCR amplification across the *ITGA4* transcript was performed using exon 1F (5′ gagagcgcgctgctttaccagg 3′) and 10R (5′ gccatcattgtcaatgtcgcca 3′) primers with the cycling conditions of 55 °C for 30 min for the reverse transcription step, followed by 28 cycles of 94 °C for 30 s, 55 °C for 30 s and 68 °C for 2 min. The RT-PCR products were fractionated on 2% agarose gels in Tris-acetate EDTA buffer and images of RedSafe™ (iNtRON Biotechnology, Inc., Burlington, MA, USA) stained gels were captured using a Fusion-FX gel documentation system (Vilber Lourmat, Marne la Vallée, France). Densitometric analysis was performed using ImageJ (NIH).

### 4.9. Protein Stress Array

The GTC PMO was delivered to fibroblasts using Endo-Porter as described above and analysis of stress proteins performed using the proteome profiler human cell stress array kit (R&D Systems, Minneapolis, MN, USA) according to the manufacturer’s protocol. Protein concentrations were determined using Pierce BCA protein assay kit (Thermo Fisher Scientific, Scoresby, VIC, Australia) and 200 µg protein was used for each assay. The pixel density was analysed using Image J (NIH).

## Figures and Tables

**Figure 1 molecules-24-02922-f001:**
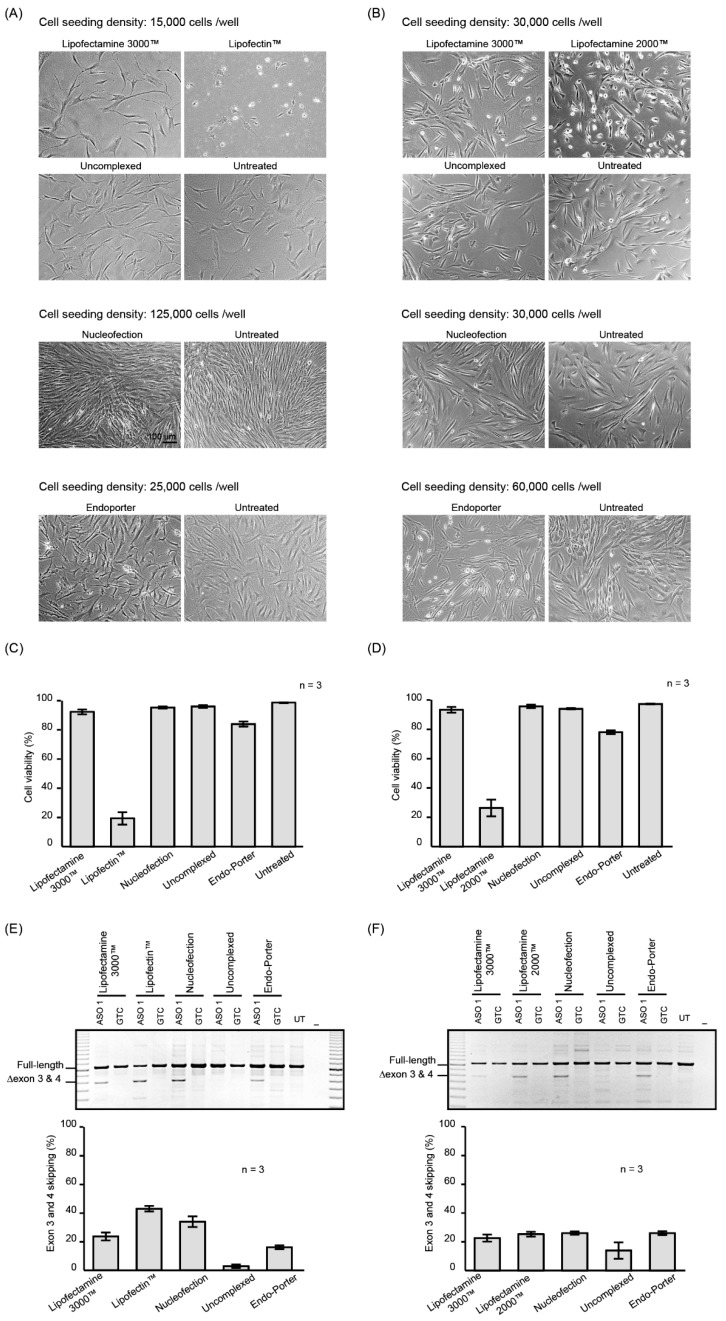
Micrographs of fibroblasts (**A**) and myoblasts (**B**) treated with 10 µM of ASO 1 for 24 h using the delivery methods indicated above each panel. Cell viability of (**C**) fibroblasts and (**D**) myoblasts treated with PMO complexed with indicated reagent, uncomplexed and untreated. RT-PCR products of the *ITGA4* transcript amplified from (**E**) fibroblasts and (**F**) myoblasts treated as in (**A**) and (**B**) respectively. Shown are the average percentages of exon skipping from three biological replicates represented as a bar graph below the gel image. GTC: Gene Tools Control PMO. Error bar; SEM.

**Figure 2 molecules-24-02922-f002:**
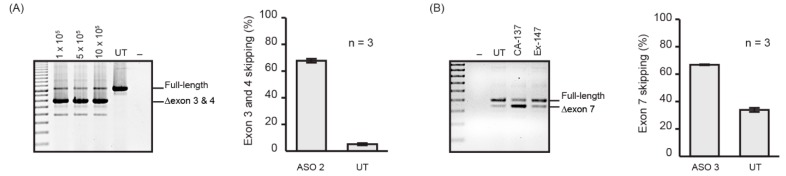
(**A**) RT-PCR products of *ITGA4* transcript amplified from human fibroblasts nucleofected with ASO 2 at 50 µM concentration in the cuvette using the P3 nucleofection kit and CA-137 program and various cell densities. The number of cells nucleofected is shown above the gel. Shown are the average percentages of exon skipping from three biological replicates represented as a bar graph next to the gel image. UT: Untreated. Error bar; SEM. (**B**) RT-PCR products of *SMN* transcript from Huh7 cells nucleofected with ASO 3 at 50 µM concentration in the cuvette using P3 kit, and CA-137 or Ex-147 program. Shown are the average percentages of exon skipping using CA-137 program from three biological replicates represented as a bar graph next to the gel image. UT: Untreated. Error bar; SEM.

**Table 1 molecules-24-02922-t001:** Optimal nucleofection programs for the P3 kit and the cell numbers tested in specified cell lines.

Cell Type	Name	P3 Program	Cell Number
Human dermal fibroblasts	HDF	CA-137	2.5−10 × 10^5^
Human skeletal muscle myoblasts	HSkM	CM-138	2.5 × 10^5^
Human T lymphocytes	Jurkat	CL-120	2-5 × 10^5^
Human Glial (Oligodendrocytic) Hybrid Cell Line	MO3.13	CA-138	2.5 × 10^5^
		DR-114	2.5 × 10^5^
		EM-110	2.5 × 10^5^
Human neuroblastoma	SH-SY5Y	CA-137	2.5−10 × 10^5^
Human hepatocarcinoma	Huh7	CA-137	2.5 × 10^5^
Human hepatocarcinoma	HepG2	EH-100	2.5 × 10^5^
Murine primary splenocytes		DN-100	10 × 10^5^
Murine myoblast	H2K*mdx*	CM-138	2.5 × 10^5^

**Table 2 molecules-24-02922-t002:** Information for ASOs.

Name	ASO Nomenclature	Sequence (5′ to 3′)	Target Protein
ASO 1	ITGA4 H3A (+ 30 + 49)	TCTCTCTCTTCCAAACAAGT	Integrin alpha 4
ASO 2	ITGA4 H3A (+ 41 + 65)	CCCCAACCACTGATTGTCTCTCTCT	Integrin alpha 4
ASO 3	Smn M7A (+ 7 + 36)	TGAGCACTTTCCTTCTTTTTTATTTTGTCT	Survival motor neuron
GTC	GTC-Gene Tools Control	CCTCTTACCTCAGTTACAATTTATA	Beta-globin chain

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
