# Peer review of "In Vitro Validation of Phosphorodiamidate Morpholino Oligomers"

_molecules, 2019, doi:10.3390/molecules24162922_

Round 1

Reviewer 1 Report

In this manuscript, the authors attempted to optimize the transfection protocols for splicing modulating PMOs targeting ITGA4 and SMN2 in several cell types. Although it is of interest to the field to see systematic comparison of different available methods to transfect PMOs, lacking convincing data seriously impairs the quality and interest of this work. Therefore, it is not recommended to publish on Molecules.

My specific major concerns are:

1. The design of this study seems random.

1) What’s the rationale to select ITGA4 and SMN2 as the targets to optimize the transfection protocols?

2) Besides electroporation, what’s the rationale to select Lipofectamine 3000™, Lipofectamine 2000™, Lipofectin™ and Endo-Porter to test? There are numerous transfection reagents available either commercially or developed by research labs, e.g. Nucleic Acids Res, 2015, 43(19), e128 showed Ca2+ enrichment in culture medium potentiates effect of oligonucleotides.

3) What’s the rationale to select these three PMOs in Table 1? The authors should mention how these sequences were screened or cite previous work.

4) In Figure 1A, 1C and 2C, how the cell seeding density is optimized? Did the authors have data for this optimization?

5) What’s the rationale to list the cell types in Figure 2C, are they all related to genetic diseases and have the potential to be the cell models for drug development?

2. To optimize the transfection protocols, transfection efficiency and cell viability are the two primary concerns. The authors should address them with quantitative comparison.

1) In Figure 1A, besides the cell morphology data, the authors should present bar graphs showing % of cell viability with different methods relative to untreated cells post 24 hours of transfection with error bars from three biological replicates.

2) In Figure 1B, 1D and 2A-B, how many biological replicates have been performed? It is suggested to present the quantitation data in bar graphs with error bars from three biological replicates.

3) In Figure 2C, what’s the transfection efficiency (% splicing switch) and cell viability after electroporation with these cells since the authors have already optimized the programs and tested PMOs in these cell models?

3. In order for research labs to use the PMOs in this work as positive transfection controls, the ASOs should have high enough potency.

1) Dose response curve with 6-7 concentrations is an unbiased way to evaluate and quantify the potency of ASOs. The authors should characterize the concentration which can cause half maximum splicing switch for the best PMO.

2) What is the maximum % splicing switch for these PMOs in cell models?

3) In this study, the authors used 10 uM for lipid/peptide mediated transfection and 50 uM for electroporation. Is it due to the low potency of the PMOs in this work since PMOs were previously reported to have nanomolar potency?

4. In Figure 1B, 1D and 2A, some smaller bands on gel were observed. Did these indicate off-target effects of ITGA4 splicing modulating PMOs?

5. In the antisense oligonucleotides field, it is suggested to use two negative controls, one mismatched control and one scrambled control, to validate any positive results. The authors should also include a scrambled control in the experiments.

Some minor points:

1. The authors should consider to abbreviate antisense oligonucleotides as “ASOs” instead of “AOs” because “ASO” is more widely used in the field.

2. On page 2 line 65, the authors wrote “previous contradicting reports...”. In vitro and in vivo data cannot be compared directly unless robust characterization of the potency was carried out both in vitro and in vivo. Therefore, “contradicting” is not a precise description here.

3. On page 3 line 84, the authors should add reference after the statement “Exondys 51, was designed in our laboratory”.

4. On page 7 line 164, in vivo delivery of synthetic nucleic acids, therapeutic vectors and plasmids is the major challenge for drug development. Consistent with the description in this manuscript, electroporation is an effective delivery approach in vitro for most cell types and is widely used in pharmaceutical industry and research labs. The authors should clarify this point.

Author Response

Response to Reviewer 1 Comments

Point 1: What’s the rationale to select ITGA4 and SMN2 as the targets to optimize the transfection protocols?

Response 1: As shown in Supplementary Figure 1, both ITGA4 and SMN2 are expressed in most cell types and cell lines that are readily available to laboratories. As such, these transcripts can be easily amplified. To emphasise this, we have now added “We have chosen previously optimised exon skipping PMOs targeting the ITGA4 and SMN transcripts expressed in most cell types and cell lines (Figure S1).” on page 4, paragraph 2.

Point 2: Besides electroporation, what’s the rationale to select Lipofectamine 3000™, Lipofectamine 2000™, Lipofectin™ and Endo-Porter to test? There are numerous transfection reagents available either commercially or developed by research labs, e.g. Nucleic Acids Res, 2015, 43(19), e128 showed Ca2+ enrichment in culture medium potentiates effect of oligonucleotides.

Response 2: As described in the introduction (page 4 paragraph 1) we are providing alternative ways to deliver PMO when electroporation systems might unavailable. The three transfection reagents we chose to test are commercially available reagents that are commonly used in most laboratories, including our own. We agree that more transfection reagents are available and different laboratories prefer to use different reagents. The main message of this report is to advise that PMO can be delivered using these reagents should the laboratory have limited or no access to an electroporation system. We are the first to report such an observation.

Point 3: What’s the rationale to select these three PMOs in Table 1? The authors should mention how these sequences were screened or cite previous work.

Response 3: These PMOs were selected based on our previous work. The Smn PMO was cited on page 8 paragraph 1 (reference [31]). We have now included the reference for the ITGA4 PMOs on page 12 paragraph 3 (reference [30]).

Point 4: In Figure 1A, 1C and 2C, how the cell seeding density is optimized? Did the authors have data for this optimization?

Response 4: The cell densities were optimised based on the manufacturer’s protocols. We have published several papers with the same seeding density shown for Figure 1, which are now added to materials and methods section 4.5 transfection protocol (reference number [31] and [36]. For the Endo-Porter experiment, we have added “80% confluency” to materials and methods section 4.5 transfection protocol. The seeding density for Figure 2C (now Table 1), is within the manufacturer’s recommendations.

Point 5: What’s the rationale to list the cell types in Figure 2C, are they all related to genetic diseases and have the potential to be the cell models for drug development?

Response 5: Type of cell and origin is provided in Table 1. These are readily and commercially available cell lines extensively used in many laboratories. To clarify this point, we have now included “that are extensively used and commercially available” in page 8, section 2.2.

Point 6: In Figure 1A, besides the cell morphology data, the authors should present bar graphs showing % of cell viability with different methods relative to untreated cells post 24 hours of transfection with error bars from three biological replicates.

Response 6: As requested, we have now added the % cell viability to Figure 1.

Point 7: In Figure 1B, 1D and 2A-B, how many biological replicates have been performed? It is suggested to present the quantitation data in bar graphs with error bars from three biological replicates.

Response 7: We have now included the quantitative data as bar graphs with error bars for three biological replicates (Figure 1).

Point 8: In Figure 2C, what’s the transfection efficiency (% splicing switch) and cell viability after electroporation with these cells since the authors have already optimized the programs and tested PMOs in these cell models?

Response 8: As we described in page 8 in section 2.2, the transfection efficiency is 80%. Cell death after electroporation/nucleofection is now added to the same paragraph as “less than 10%”.

Point 9: P4/l.107: Dose response curve with 6-7 concentrations is an unbiased way to evaluate and quantify the potency of ASOs. The authors should characterize the concentration which can cause half maximum splicing switch for the best PMO.

Response 9: The dose response for these ASOs have been published in references [31] and [30]. The purpose of this report is not to identify the best ASO but rather to indicate different ways of delivery.

Point 10: In this study, the authors used 10 uM for lipid/peptide mediated transfection and 50 uM for electroporation. Is it due to the low potency of the PMOs in this work since PMOs were previously reported to have nanomolar potency?

Response 10: To this day there has been no report of an unconjugated PMO working at a nanomolar concentration. The reviewer may have misunderstood that all antisense are the same. Although siRNA, RNase H dependent antisense may be active at nanomolar concentrations, PMO on the other hand requires a micromolar concentration. The concentrations reported at 10 μM are on the low end of what is recommended for transfection reagent Endo-Porter.

For nucleofection, 50 μM was used due to high number of cells (Figure 2 and Table 1). In addition, we report the nucleofection concentration in the cuvette rather than final seeding volume. If we reported Nucleofection concentration in final seeding volume of 1 ml, then the concentration would be lower, at 1 µM. However, as final seeding volume can be interchangeable, we prefer to use concentration within the cuvette for consistency and accuracy.

Point 11: In Figure 1B, 1D and 2A, some smaller bands on gel were observed. Did these indicate off-target effects of ITGA4 splicing modulating PMOs?

Response 11: These smaller bands have been identified as alternative transcripts of ITGA4. ITGA4, like most gene transcripts, undergoes natural splice modulation giving rise to various (11 reported to date) alternative transcripts. Most of the smaller bands observed are also evident in the untreated sample.

For Reviewer’s information: (http://asia.ensembl.org/Homo_sapiens/Gene/Splice?db=core;g=ENSG00000115232;r=2:181457202-181538940)

Point 12: In the antisense oligonucleotides field, it is suggested to use two negative controls, one mismatched control and one scrambled control, to validate any positive results. The authors should also include a scrambled control in the experiments.

Response 12: As we are only assessing splice modulation rather than knockdown, the presence of shorter transcripts in addition to the full length shows the effectiveness of the ASO. In addition, the Gene Tools Standard Control oligo only targets reticulocytes from individuals with thalassemia having a splice-generating mutation at position 705 in beta-globin pre-mRNA. One control should therefore be adequate for these methods. Numerous in vitro and in vivo citations with one control oligo can be provided at reviewer’s request.

Some minor points:

Point 13: The authors should consider to abbreviate antisense oligonucleotides as “ASOs” instead of “AOs” because “ASO” is more widely used in the field.

Response 13: “AOs” are now amended as “ASO” throughout the text as suggested by reviewer.

Point 14: On page 2 line 65, the authors wrote “previous contradicting reports...”. In vitro and in vivo data cannot be compared directly unless robust characterization of the potency was carried out both in vitro and in vivo. Therefore, “contradicting” is not a precise description here.

Response 14: The word “contradicting” has been removed (page 3 paragraph 2).

Point 15: On page 3 line 84, the authors should add reference after the statement “Exondys 51, was designed in our laboratory”.

Response 15: References 25-27 are cited as they are the most published data on Eteplirsen. Professors Wilton and Fletcher have been attributed to the development of Eteplirsen and we have now included into references the drug patent [24] (page 3 paragraph 5) and the conflict of interest statement.  

Point 16: On page 7 line 164, in vivo delivery of synthetic nucleic acids, therapeutic vectors and plasmids is the major challenge for drug development. Consistent with the description in this manuscript, electroporation is an effective delivery approach in vitro for most cell types and is widely used in pharmaceutical industry and research labs. The authors should clarify this point.

Response 16: We have now reworded as “While clearly the design of the therapeutic vector, plasmid or antisense oligomer is a fundamentally crucial aspect, without efficient and effective delivery to the target tissue or cells, the most therapeutically active compound may appear inert” on page 10 paragraph 1.

Reviewer 2 Report

The manuscript of Aung-Htut et. al. reports a comparative study of various approaches to deliver phosphorodiamidate morpholino oligonucleotides (PMOs) into cells. PMOs are attractive as antisense and splice-switching oligonucleotides but suffer from poor delivery properties. The topic is timely and ideally suited for the special issue “Antisense Oligonucleotide Chemistry and Applications”. The manuscript is well written, the experimental work carried out to a high standard and the conclusions supported by the data obtained. The field will certainly benefit from the rigorous comparison of the various delivery methods. I therefore recommend publication in Molecules and only have the following minor concerns:

Both the abstract and the discussion contain text (lines 20 – 27 and 164 – 196) that would better fit under introduction.

On line 59, the authors (correctly) state that “PMOs have a neutral charge and delivering these compounds into cultured cells can be problematic.” On the other hand, unmodified oligonucleotides also do not penetrate into cells easily and this is generally attributed to their polyanionic charge. I understand that PMOs would not interact with cationic lipids as strongly as negatively charged oligonucleotides but the fact that the PMOs also show poor gymnotic uptake is less obvious. Could the reasons behind the difficulties in delivering PMOs to cells be elaborated?

As figure 2C is essentially a table, it should be presented as such, rather than as part of a figure.

Author Response

Response to Reviewer 2 Comments

Point 1: Both the abstract and the discussion contain text (lines 20 – 27 and 164 – 196) that would better fit under introduction.

Response 1: We have now removed and reworded. Some sentences added to the introduction (page 2 paragraph 1 and 3; page 3 paragraph 3).

Point 2: On line 59, the authors (correctly) state that “PMOs have a neutral charge and delivering these compounds into cultured cells can be problematic.” On the other hand, unmodified oligonucleotides also do not penetrate into cells easily and this is generally attributed to their polyanionic charge. I understand that PMOs would not interact with cationic lipids as strongly as negatively charged oligonucleotides but the fact that the PMOs also show poor gymnotic uptake is less obvious. Could the reasons behind the difficulties in delivering PMOs to cells be elaborated?

Response 2: As requested, we have now provided elaboration (page 2 paragraph 3; page 3 paragraph 2)

Point 3: As figure 2C is essentially a table, it should be presented as such, rather than as part of a figure.

Response 3: We have now removed Figure 2C from Figure 2 and created a standalone Table (Table 1).

Reviewer 3 Report

The paper by Aung-Htut et al. describes the results of systematic studies on the efficiency of PMO oligomers delivery  in a series of human and mouse cells in the presence of selected transfection agents. The authors tested three lipid-based reagents (Lipofectamine 3000, Lipofectamine 2000 and Lipofectin) as well as the Endo-Porter reagent. Besides, they tested the electroporation method alone. The applicability of the nucleofection and investigated transfecting agents was tested in the exon skipping system, and the presence of the expected mRNA product was assessed by an RT-PCR approach. The methods used are rather routine, but the results give the readers an idea about the efficiency and cytotoxicity of the given transfection approach for the cellular system used.

Needed corrections/improvements:

Please,  use properly the term “sequences”, which  is the order of nucleotides in an oligonucleotide chain (usually  written from the 5’-end). Do not use “sequences” as an alternative for oligomers/candidates etc. Correct along the text.

Replace the abbreviation  “AOs” by “ASOs” (antisense oligonucleotides), the latter is commonly used in the literature.

Describe shortly the Endo-Porter reagent

P2/l.43: replace the term “13 base oligodeoxyribonucleotide”with: “13 –nucleotide (nt) oligomer”

Along the text change “2OMe” into “2’-OMe units” or “2’-OMe-backbone”

The references 22-24 refer to “own work”, however, none of their authors can be found in the author’s list of the reviewed manuscript. Ref 22 is not present in the WoS database!

Please, clarify the expression: p3/l.84 (we designed); Reword if necessary;

The names for ASOs are rather complicated and could be simplified to 1,2 and 3, and explained in Table 1.

P4/l.107: It seems that the meaning is not correctly expressed. Please, replace “…the alternative transfection strategies of nucleofection,…” for “…the alternative for nucleofection transfection strategies,…”

P4/l.116 “somewhat interesting” ? reword

Figure 1. The annotation should be simplified (remove the name of PMO and give a symbol 1.

Figure 2. the description of Fig. 2a is incorrect (should be 1 , 5 and 10 x 105)

P.7/l.164: the first sentence in Discussion is not clear (gene and genetic therapies??). Please, reword.

P.10/l. 258-259 reword (no verb is given)

Add the reference on PMO uptake: Shouta Miyatake et al Molecular Therapy - Nucleic Acids, March 2019

Author Response

Response to Reviewer 3 Comments

Point 1: Please, use properly the term “sequences”, which is the order of nucleotides in an oligonucleotide chain (usually written from the 5’-end). Do not use “sequences” as an alternative for oligomers/candidates etc. Correct along the text.

Response 1: As recommended, we have now replaced the “sequences” with “oligomers” as requested.

Point 2: Replace the abbreviation “AOs” by “ASOs” (antisense oligonucleotides), the latter is commonly used in the literature.

Response 2: We have now replaced all “AO” abbreviations with “ASO” throughout the paper.

Point 3: Describe shortly the Endo-Porter reagent

Response 3: We have now added “Gene Tools has developed a novel peptide-based delivery reagent, Endo-Porter, that allows PMO delivery via endocytosis to adhered cells without the use of a leash [21]” (page 3 paragraph 4).

Point 4: P2/l.43: replace the term “13 base oligodeoxyribonucleotide”with: “13 –nucleotide (nt) oligomer”

Response 4: We have changed the description to 13 –nucleotide (nt) oligomer”.

Point 5: Along the text change “2OMe” into “2’-OMe units” or “2’-OMe-backbone”

Response 5: We have now replaced the text 2OMe into “2’-OMe” throughout the article.

Point 6: The references 22-24 (23-25) refer to “own work”, however, none of their authors can be found in the author’s list of the reviewed manuscript. Ref 22 is not present in the WoS database!

Response 6: References [25-27] are cited as they are the most published data on Eteplirsen. Professors Wilton and Fletcher have been attributed to the development of Eteplirsen and we have now included into references the drug patent (Wilton, Stephen Donald, Sue Fletcher, and Graham McClorey. "Antisense oligonucleotides for inducing exon skipping and methods of use thereof." U.S. Patent 7,960,541, issued June 14, 2011) as reference [24] and author listed manuscript.  

Reference [22] is found on the NIH and is a book section, we are unsure as to why it is not present on the WoS database, URL provided for referral (https://www.ncbi.nlm.nih.gov/pmc/articles/PMC6195309/)

Point 7: Please, clarify the expression: p3/l.84 (we designed); Reword if necessary;

Response 7: “We designed” means the authors (Wilton and Fletcher) designed and tested the 5’-3’ sequence that anneals to exon 51 of the DMD transcript.

We have now changed the word “designed” to “developed” page 3 paragraph 5.

Point 8: The names for ASOs are rather complicated and could be simplified to 1,2 and 3, and explained in Table 1.

Response 8: We have now amended to ASO 1, 2, 3 and added to Table 2.

Point 9: P4/l.107: It seems that the meaning is not correctly expressed. Please, replace “…the alternative transfection strategies of nucleofection,…” for “…the alternative for nucleofection transfection strategies,…”

Response 9: We have now replaced “…the alternative transfection strategies of nucleofection,…” to  “the alternative transfection strategies such as nucleofection…” (page 4 paragraph 2).

Point 10: P4/l.116 “somewhat interesting” ? reword

Response 10: We have now amended “somewhat interesting” to “atypical when compared to other reagents” (page 5 paragraph 1).

Point 11: Figure 1. The annotation should be simplified (remove the name of PMO and give a symbol 1.

Response 11: We have now amended to ASO 1 in Figure 1.

Point 12: Figure 2. the description of Fig. 2a is incorrect (should be 1 , 5 and 10 x 105)

Response 12: We have now amended to 10 x 105.

Point 13: P.7/l.164: the first sentence in Discussion is not clear (gene and genetic therapies??). Please, reword.

Response 13: We have now removed “gene and” (page 10 paragraph 1).

Point 14: P.10/l. 258-259 reword (no verb is given).

Response 14: We have now amended to “These oligomers target ubiquitously expressed genes (ITGA4 and Smn) and may be useful as positive transfection controls in other studies.” (page 12 paragraph 4)

Point 15: Add the reference on PMO uptake: Shouta Miyatake et al Molecular Therapy - Nucleic Acids, March 2019

Response 15: We thank the reviewer for this information and have now added this reference [15] (page 2 paragraph 3; page 3 paragraph 1).

Round 2

Reviewer 1 Report

The authors have addressed most of my concerns in their rebuttal letter. There is only one minor revision now.

1. Similar to Figure 1E-F, bar graph quantification with three biological replicates should be added for Figure 2A-B.

Author Response

Point 1: 1. Similar to Figure 1E-F, bar graph quantification with three biological replicates should be added for Figure 2A-B.

Response 1: Included bar graph with error bars from three biological replicates in Figure 2A-B. (page 8; Line 189)